# Cardiac Function and Fatigue During Exoskeleton-Assisted Sit-to-Stand Maneuver and Walking in People with Stroke with Moderate to Severe Gait Disability: A Pilot Cross-Sectional Study

**DOI:** 10.3390/s25010172

**Published:** 2024-12-31

**Authors:** Raimondas Kubilius, Darius Ruočkus, Vitalija Stonkuvienė, Rugilė Vareikaitė, Rebecca Cardini, Thomas Bowman

**Affiliations:** 1Department of Physical Rehabilitation Medicine, Lithuania University of Health Sciences (LUHS), 50161 Kaunas, Lithuania; dariusruockus1@gmail.com (D.R.); vitalijabarasaite@gmail.com (V.S.); rvareikait@gmail.com (R.V.); 2Department of Pathophysiology and Transplantation, University of Milan, 20100 Milan, Italy; rcardini@dongnocchi.it; 3IRCCS Fondazione Don Carlo Gnocchi, 20148 Milan, Italy; tbowman@dongnocchi.it

**Keywords:** stroke rehabilitation, robotic-assisted gait training, exoskeleton, cardiac function, fatigability

## Abstract

Background. Wearable powered exoskeletons could be used to provide robotic-assisted gait training (RAGT) in people with stroke (PwST) and walking disability. The study aims to compare the differences in cardiac function, fatigue, and workload during activities of daily living (ADLs), while wearing an exoskeleton. Methods. Five PwST were recruited in this pilot cross-sectional study. We observed three experimental conditions: walking without and with the UAN.GO exoskeleton and walking with the UAN.GO combined with the OPTIGO walker. Each condition included five trials related to ADLs such as sitting and walking. Results. No statistically significant difference was found between heart rate and R–R of ECG data while comparing all the observed conditions during each respective trial. The NASA Task Load Index did not show significant differences across all trials, except for a significant difference between Condition 2 and Condition 3 in Trial 4 (*p* = 0.043). However, walking and sit-to-stand tasks seem to be more challenging according to the NASA-TLX. Only one participant scored over 70 points on the System Usability Scale. The TSQ-WT scores for conditions 2 and 3 were 62 (56.5–72.5) and 70 (66.5–75) points, respectively. Conclusions. This study suggests that UAN.GO exoskeleton could be used for RAGT in PwST with disability without compromising cardiovascular function.

## 1. Introduction

Stroke is one of the leading causes of disability worldwide with an increasing prevalence rate in recent years [1]. Several neurological impairments affect People with Stroke (PwST), such as hemiparesis, which often compromises the subject’s Activities of Daily Living (ADLs), including walking [2]. Indeed, one-third of patients with stroke do not regain independent walking ability. For those who do, their gait is mainly characterized by a decreased walking speed, increased stride width, and double support phase, leading to an asymmetrical pattern [3].

After a stroke, the long-term goal of rehabilitation is for the subjects to be successfully reintegrated into the community, with the recovery of walking ability as the top functional objective. To reach this objective, stroke rehabilitation programs should include meaningful, repetitive, intensive, and task-specific movement training in an enriched environment to promote neural plasticity and motor recovery [3].

Various novel stroke rehabilitation techniques for motor recovery have been developed based on basic science, clinical studies of neural plasticity, and exploiting the advantage of technologies [3]. Among them, robotic-assisted gait training (RAGT) could be promoted as a novel technique capable of improving walking ability among PwST [4]. Indeed, RAGT has shown clinically significant improvements in gait and balance outcomes and could be considered a valid approach to enhance gait function in people with severe gait impairments due to neurological diseases [5].

Even though there is no clear evidence, it seems that the combination of RAGT and conventional treatment could lead to better outcomes than the two techniques performed singly [6]. Therefore, on a clinical level, RAGT and conventional treatment should be performed together when both are available in the case of post-stroke subjects with balance impairment [7]. Although the combination of RAGT and conventional treatments has proven to be effective for motor recovery, conventional overground walking training can be very difficult or even impossible for PwST with moderate to severe gait disability and limited cardiovascular capacity due to too high energy demand [8]. Moreover, people with severe gait impairments can suffer from disturbed cardiac function during exercise which affects exercise tolerance and balance during orthostatic challenges, such as the sit-to-stand maneuver. Indeed, altered cardiac function and fatigability are common phenomena in stroke survivors and wearable powered exoskeletons could be used to provide RAGT safely [9]. Powered exoskeletons may increase energy expenditure to a similar level as non-exoskeleton walking, which may improve cardiovascular function more effectively than wheelchair propulsion alone [10]. Moreover, cardiorespiratory and metabolic demands of exoskeleton-assisted walking overground should be consistent with physical activities performed at a moderate intensity. Prolonged bouts of exoskeleton-assisted walking may provide a stimulus sufficient to improve cardiorespiratory fitness [11].

However, few studies provided preliminary evidence on the impact of RAGT provided with an exoskeleton on cardiovascular health, energy expenditure, body composition, gait parameters, level of physical activity, and quality of life in PwST [12].

Before recommending RAGT in PwST with moderate to severe gait impairments, it is necessary to provide additional cardiovascular parameters and fatigability data comparing activities such as sit-to-stand and walking performed with and without an exoskeleton.

In fact, exoskeleton devices could be used to promote prolonged walking with acceptable levels of fatigue in severely disabled people by allowing safe stimulation of cardiovascular parameters to improve cardiovascular function. According to this, we hypothesized that RAGT has the potential to ease the process of gait training while not aggravating cardiovascular demand for PwST.

Therefore, the study aims to measure cardiac function, fatigability, and workload during ADLs, such as getting up from a chair and walking, while wearing an electrically powered exoskeleton in different support modes compared to the same ADLs performed without an exoskeleton in PwST with moderate to severe walking impairment. Moreover, we want to investigate the usability of the exoskeleton.

## 2. Materials and Methods

### 2.1. Participants and Entry Criteria

This pilot study followed a cross-sectional design involving PwST at the LUHS Kulautuva Hospital of Rehabilitation from 28 December 2022 to 28 February 2023.

Inclusion criteria were diagnosis of stroke (1 ≤ Functional Ambulation Category (FAC) ≤ 3), age > 18, able to perform 4 min of back-and-forth locomotion with assistance or resting periods, height between 160 and 195 cm, and weight < 100 kg.

Exclusion criteria were impairments in the upper limbs that do not allow the patient to hold the crutches/walker, skin injuries in the areas where the exoskeleton is in contact with the user, fractures not solved or bone pathologies in lower limbs in which the use of the exoskeleton could be risky (advanced osteoporosis), psychiatric or cognitive problems that can interfere with the correct use of the device, important muscle/joint retractions in lower limbs (Modified Ashworth Scale > 3), any medical conditions that could interfere with cardiac function, e.g., severe diabetes, use of beta-blocker drugs, and the presence of severe cardiovascular diseases.

### 2.2. Equipment

UAN.GO exoskeleton (Figure 1)—a medical device, CE class IIA certified. UAN.GO (U&O, 29017 Fiorenzuola d’Arda, Italy) is a powered exoskeleton for lower limb disabilities and gait impairments. It is a medical device for overground RAGT. The exoskeleton is equipped with 8 kinematics joints: 4 mechatronics joints (hips and knees), 2 passively actuated joints with spring system (ankles), and 2 passive joints (feet). Different ranges of motions are available for the hip (100° flexion, 30° extension), knee (110° flexion, 0° extension), and ankle (0° plantarflexion, 10° dorsiflexion). To control the exoskeleton, an operator selects the assistive modalities. In our study we only use 2 modalities: (1) passive, when the movement is 100% assisted by the exoskeleton and guided for safety by the operator, and (2) active-assisted, when the movement is 75% assisted by the exoskeleton and partial movement effort of the subject is needed (also guided and assisted for safety by the operator);OPTIGO walker—a medical device, CE class IIA certified (U&O, 29017 Fiorenzuola d’Arda, Italy): this is a walker supporting the subject in the sit-to-stand and walking operation integrating a wired remote control for subject lift. Device characteristics are weight ~20 kg, dimensions (length × width × height) 135 × 82 × 122 cm. It is possible to combine the OPTIGO walker with the UAN.GO exoskeleton to obtain the UAN.GO–OPTIGO platform (Figure 2);Polar H9 with Polar Soft Strap (Polar Electro Oy, Kempele, Finland)—This is a medical device containing a sensor to measure heart rate and R–R interval.

### 2.3. Experimental Procedures

Once the eligible subjects were selected according to the inclusion and exclusion criteria and the informed consent forms were signed, participants performed the experimental procedures.

The experimental procedures comprised 5 experimental sessions (Table 1) performed on different and consecutive days. Sessions 1 and 2 consisted of a practice period of 30 min to help subjects adapt to the exoskeleton and UAN.GO–OPTIGO walker platform. During these practice sessions, subjects performed sitting, standing, and guided walkthroughs with UAN.GO exoskeleton helped by a physiotherapist. The same tasks were repeated in the second practice session using the UAN.GO–OPTIGO platform. In the other 3 sessions, each subject performed 3 experimental conditions as shown in Table 1 (Condition 1: Walking without exoskeleton; Condition 2: Walking with UAN.GO exoskeleton; Condition 3: Walking with exoskeleton and walker—UAN.GO–OPTIGO platform). The order of days on which a subject performed Condition 1, Condition 2, or Condition 3 was randomized. Each condition consisted of 5 trials: Trial 1—Sitting for 5 min in a comfortable position; Trial 2—Standing for 5 min with assistance; Trial 3—Walking at a comfortable speed (100% assisted) for 4 min on a 20-m linear flat path; Trial 4—Walking at a comfortable speed (75% assisted) for 4 min on a 20-m linear flat path (not available in Condition 1); and Trial 5—1 min Sit-to-Stand test.

### 2.4. Outcome Measures and Evaluation

The following outcome measures have been used to describe the clinical characteristics of the participants:

Functional Independence Measure (FIM)—a test that assesses a person’s level of independence in various areas of daily living. It evaluates self-care, sphincter control, transfers, mobility, communication, and social cognition. An 18-item, seven-level ordinal scale designed to detect changes in functional status throughout a comprehensive inpatient medical rehabilitation program. It evaluates the level of assistance required by an individual, ranging from total independence to total assistance [13].

Barthel Index—an ordinal scale that evaluates an individual’s ability to perform activities of daily living (ADLs). The score can vary from to 0 to 100 [14]. In LUHS Kulautuva rehabilitation hospital, the PM&R physician evaluates patients’ ADLs and determines the score accordingly before the admission to inpatient rehabilitation facility and at the end rehabilitation period before the discharge from the facility.

Manual Muscle Testing was evaluated according to the Lovett scale. This scale is a commonly used tool by doctors and therapists to assess a patient’s muscle strength. The scale assigns a score between 0 and 5 to each muscle group in the limbs: 0—no movement; 1—contraction visible or palpable, fasciculations; 2—active movement with gravity eliminated; 3—active movement against gravity; 4—active muscle contraction against gravity with some resistance; 5—active muscle concentration against full resistance; full strength [15].

The following outcome measures have been used to assess cardiovascular function:-Heart Rate (HR) and R–R interval: every second, the Polar H9 transmits a data packet containing one averaged HR value (derived from a circular buffer) and one or more R–R intervals measured in real time between consecutive heartbeats recorded in the previous second. This results in an evenly spaced HR time series and a non-uniformly spaced R–R interval time series. The HR is the number of heartbeats per unit of time. Usually expressed as beats per minute. The R–R interval is the distance between two consecutives R waves. The duration of the R–R interval depends on the HR and it is usually expressed in milliseconds (ms) [16].

The following outcome measure has been used to assess fatigue and workload:-The NASA Task Load Index (NASA-TLX) is a subjective, self-assessment instrument. Users rate their experience of the exercise/task on six aspects, as a point on a 20-unit scale: mental demand, physical demand, temporal demand, performance, effort, and emotional stress. These are combined to form a single index [17]. NASA-TLX was requested at the end of each trial.

The following outcome measures have been used to assess usability and satisfaction:-System Usability Scale (SUS). The SUS is a simple 5-point Likert scale providing a global view of subjective usability ratings, developed as a quick and efficient way to obtain an overview of system usability. As an advantage, the SUS is quick and easy to use for both participants and researchers because provides a single score on an easy-to-understand scale. The SUS consists of 10 questions. The participant’s scores for each question are converted to a new number, added together, and then multiplied by 2.5 to convert the original scores of 0–40 to 0–100 [18]. Usability questionnaires were completed at the end of Condition 2 and Condition 3.-The Tele-healthcare Satisfaction Questionnaire—Wearable Technology (TSQ-WT) is a questionnaire developed to measure the satisfaction of users about using wearable technologies. It consists of six dimensions evaluating the benefit, usability, self-concept, privacy and loss of control, quality of life, and wearing comfort of a system. Each dimension includes five items rated on a 5-point scale with higher scores indicating more positive ratings [19]. TSQ-WT was used to assess benefit, usability, self-concept, privacy and loss of control, and wearing comfort from 0 (not at all) to 4 (fully agree). Total scores range from 0 to 120 [20]. Usability questionnaires were completed at the end of Condition 2 and Condition 3.

## 3. Statistical Analysis

Statistical analysis was performed using MS Excel and the software package IBM SPPS Statistics 29 for Windows. Data were checked for normality and presented as mean (SD) or median (1°Q–3°Q) according to their distribution. All values were rounded to two decimal places.

Considering cardiovascular function, HR and R–R interval were extracted and filtered from the Polar H9 device and all signal artefacts were removed. Means of HR and R–R values from each subject (N = 5) between different conditions during each trial were tested for distribution properties. Because the sample size is low (N = 5), Shapiro–Wilk criteria were used. Requirements for normality (Gauss) distribution were met (when *p* > 0.05 in the test):
(a)HR means data:
Trial 1 under Condition 1; Trial 1 under Condition 3;Trial 2 under Condition 1; Trial 2 under Condition 3;Trial 3 under Condition 1; Trial 3 under Condition 2;Trial 4 under Condition 2; Trial 4 under Condition 3;Trial 5 under Condition 2; Trial 5 under Condition 3.
(b)R–R means data:
Trial 3 under Condition 1; Trial 3 under Condition 2;Trial 4 under Condition 2.


For comparison of data combinations mentioned in the bullets above, a paired samples T test was used.

Yet again, because of the low sample size even for normally distributed data, to compare these data, it was recommended to use non-parametric criteria—Kruskal–Wallis.

For the remaining trial and condition combinations, the data requirements for normal (Gauss) distribution were not met (*p* < 0.05 in the test). For these data, Kruskal–Wallis criteria for two dependent samples were used.

Considering the fatigue and workload questionnaire, a non-parametric K-related Friedman test was used. The NASA-TLX total score was calculated using the test‘s raw score according to the literature [21]. A non-parametric K-related Friedman test was used to compare NASA-TLX scores of each trial among all conditions. Since Trial 4 does not exist for Condition 1, we compared Trial 4 NASA-TLX scores between conditions 2 and 3 using the non-parametric Wilcoxon test for two related samples.

Considering the usability and satisfaction assessments (TSQ-WT and SUS), for each questionnaire a non-parametric Wilcoxon test was used to compare trials between conditions 2 and 3.

Results are interpreted as statistically significant when *p* ≤ 0.05.

## 4. Results

Five PwST (four males and one female) with a mean (SD) age of 72.6 (4.6) years were eligible and agreed to participate in the study. The baseline characteristics are described in Table 2.

### 4.1. Cardiovascular Function

In this section, HR and R–R interval data are shown, respectively, in Table 3 and Table 4.

Considering the HR analysis of cardiovascular function (Table 3), no statistically significant differences were found between trials among conditions 1, 2, and 3 in all comparisons except for two. These are: Trial 5, Condition 1 vs. Condition 2 (*p* = 0.043) and Trial 5, Condition 1 vs. Condition 3 (*p* = 0.043).

Considering the R–R interval analysis of cardiovascular function (Table 4), no statistically significant differences were found between conditions 1, 2, and 3 among all trials in all comparisons except for two. These are: Trial 5, Condition 1 vs. Condition 2 (*p* = 0.043), and Trial 5, Condition 1 vs. Condition 3 (*p* = 0.043).

### 4.2. Fatigue and Workload

According to the NASA-TLX (Table 5), the overall statistical analysis revealed no significant difference in each trial among all conditions. Considering Trial 4, we observed a statistically significant difference in NASA-TLX scores between Conditions 2 and 3 (*p* = 0.043).

### 4.3. System Usability and Satisfaction

Table 6 shows the TSQ-WT questionnaire scores of each subject. Considering Condition 2, the median (1Q–3Q) of the whole sample is 62 (56.5–72.5) points while considering Condition 3 is 70 (66.5–75) points. No statistically significant difference was found between the TSQ-WT total scores of Conditions 2 and 3.

Table 7 shows the SUS scores. Considering Condition 2, the median (1Q–3Q) of the whole sample is 35 (26.25–40) points. Considering Condition 3, the median (1Q–3Q) of the whole sample is 47.5 (43.75–60) points. The statistical comparison reveals that SUS scores showed a statistically significant difference between Condition 2 and Condition 3 (*p* = 0.043).

## 5. Discussion

In this study, we investigated the differences in cardiac function and fatigue during ADLs while wearing an electrically powered exoskeleton in different support modes, compared to the same ADLs performed without an exoskeleton in PwST. We also investigated the usability of the exoskeleton.

Five PwST with severe walking disability were recruited and completed the study. Before the study, all participants received initial treatment in the Neurology department and underwent initial rehabilitation, including physical therapy and neuromuscular electrical stimulation (NMES). Subsequently, these patients were admitted to LUHS Kulautuva Hospital of Rehabilitation for the second stage of inpatient rehabilitation. During their second stay, they received conventional rehabilitation treatment, which encompassed physical therapy, occupational therapy, and various physical therapy techniques such as NMES, ultrasound therapy, lymph drainage, and magnetotherapy. Additionally, they received massage, video relaxation therapy, and interventions from psychology and social work specialists. In this context, participants were involved in the experimentation.

In general, people receiving RAGT in combination with physiotherapy after stroke are more likely to achieve independent walking than people who receive gait training without these devices, as reported by the literature [22]. Even though exoskeleton assistance is seen as a valuable aid for gait rehabilitation process of people with walking disabilities, we need to explore how challenging it is in terms of cardiovascular demands, perceived fatigue, and usability, especially in subjects with moderate to severe walking disability. Since poor gait and endurance limit daily walking in stroke patients [23], we follow the hypothesis that RAGT could aid gait training without adding cardiovascular strain in PwST. Our results revealed no significant differences in cardiovascular function when comparing data across all three conditions. These results may suggest that RAGT with an exoskeleton could be used to provide longer and continuous walking training without compromising cardiovascular function in PwST with moderate–severe walking disability. Since prolonged, constant moderate-intensity exercise, performed while maintaining control of cardiac function, is the foundation of cardiovascular reconditioning, RAGT could be used to promote improvements in gait and cardiopulmonary fitness compared to traditional gait training for PwST [24]. Indeed, a previous study suggested that RAGT showed the potential to improve cardiopulmonary fitness after stroke, even for subjects who are in the early stages of recovery and not independent ambulators [25]. Similar considerations about exoskeleton-assisted walking have been found even for other neurological populations, such as motor-complete spinal cord injuries, in which the energy expenditure was lower with exoskeleton-assisted walking than non-exoskeleton walking and the rate of perceived exertion reported during exoskeleton-assisted walking was equal to moderate intensity [11].

When considering fatigue, we found conflicting results when assessing perceived fatigue after a single task and the overall fatigue at the end of each experimental condition. All participants, except subject 4, showed a higher level of perceived fatigue while walking at a comfortable speed and during the sit-to-stand task using the exoskeleton. In this assessment, the exoskeleton-assisted condition without a walker resulted as the most demanding. These results might indicate that the single walking task performed with an exoskeleton may induce a moderate-intensity effort. In fact, even when partially or fully assisted by the robot, participants still need to engage with muscle activation in the lower limbs and trunk, which often differs from the normal muscular activation during non-robot-assisted walking [26].

Finally, the usability assessment revealed that most of the subjects reported poor usability of the exoskeleton in both conditions according to the SUS. Only subject 3 reported “good” usability when using the UAN.GO–OPTIGO platform. On the other hand, when using the TSQ-WT to measure user satisfaction we discovered that the average scores in both conditions are in the upper third quartile, which could be interpreted as a “moderate” to “good” satisfaction of the device usability. The differences in the usability assessment could be due to the fact that the SUS scale is used as a quick evaluation tool and contains only 10 items, which poses the risk of losing some information. In contrast, the TSQ-WT measures various aspects related to the use of the device, particularly benefit, usability, self-concept, privacy and loss of control, quality of life, and wearing comfort, with a total of 30 items [19]. The latter might be a more reliable tool than the former for providing a comprehensive evaluation of a device’s usability. However, based on the subjects’ experience, all of them commented that using the UAN.GO–OPTIGO platform they felt more accommodating and safer than only walking with the exoskeleton due to the fear of falling. Future clinical practice should consider the balance between actual and perceived benefits, as well as the potential barriers to integrating an exoskeleton into stroke rehabilitation [27].

The main limitation of our study is the small sample size. This was primarily due to the short duration of the research, as the exoskeleton was only available at our hospital for a limited time. Given this constraint, further research with a larger sample size is necessary. Additional studies with significantly more participants are required to validate the hypotheses and findings discussed here. Moreover, while we measured cardiovascular parameters such as HR and R–R intervals, we were unfortunately unable to assess key cardiopulmonary variables like VO_2_ max, which is essential for evaluating cardiopulmonary fitness. This limits our ability to fully assess the exoskeleton’s impact on cardiovascular demand.

## 6. Conclusions

The findings of this pilot study suggest that an exoskeletal robotic device may be a viable option for gait training in people with moderate to severe disability, as it does not appear to compromise cardiovascular function. In addition, data analysis showed that during Trial 5 (sit-stand for a 1 min task) subjects under Condition 2 (using only the exoskeleton) and Condition 3 (using the exoskeleton and OPTIGO walker) had less cardiovascular stress than performing this trial under Condition 1 (without the exoskeleton). This could possibly show that depending on the task using the exoskeleton could even lower the cardiovascular function demand.

However, further studies with larger sample sizes are required to confirm these preliminary results. Overall, subjects described the combination of the exoskeleton with a walker (UAN.GO–OPTIGO platform) as more usable and less challenging if compared to the exoskeleton alone. However, the device’s usability aspects should be improved based on the evaluation of the participants.

## Figures and Tables

**Figure 1 sensors-25-00172-f001:**
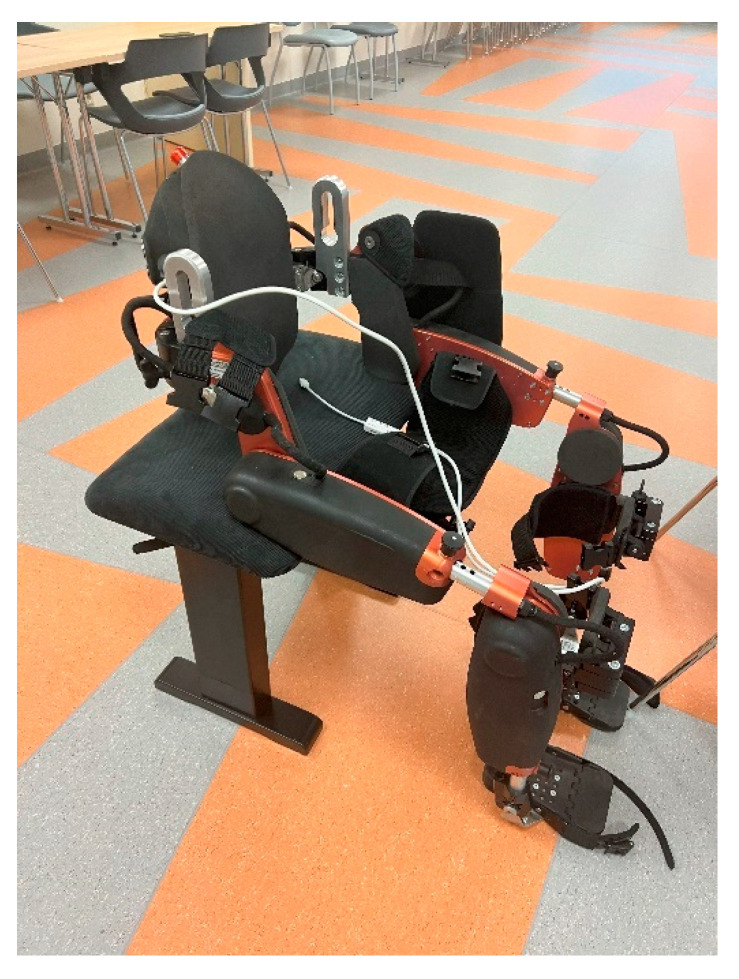
UAN.GO exoskeleton.

**Figure 2 sensors-25-00172-f002:**
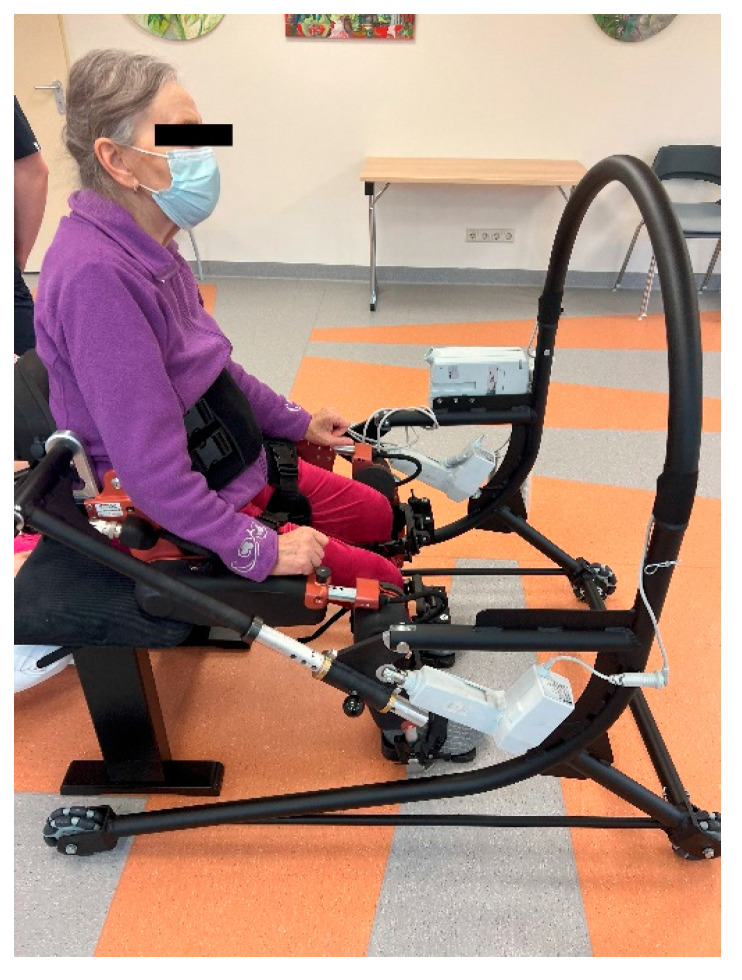
UAN.GO–OPTIGO platform.

**Table 1 sensors-25-00172-t001:** Experimental procedures.

Preparation Phase		Experimental Phase		
Screening	Exo Training	Condition 1	Condition 2	Condition 3
		Trials without Exoskeleton	Trials with UAN.GO Exoskeleton	Trials with Exoskeleton and Walker (UAN.GO–OPTIGO platform)
Session 0	Sessions 1–2	Session 3	Session 4	Session 5
Screening Test and informed consent form	-A practice period will be provided to help subjects adapt to the wearable robot	Trial 1—Sitting: 5 min, sitting at rest in comfortable position	Trial 1—Sitting: 5 min, sitting at rest in comfortable position	Trial 1—Sitting: 5 min, sitting at rest in comfortable position
		Trial 2—Standing: 5 min, with assistance	Trial 2—Standing: 5 min, with assistance	Trial 2—Standing: 5 min, with assistance
		Trial 3—Comfortable speed (CS) Walking: back-and-forth locomotion for 4 min on a 20 m length linear flat path with assistance	Trial 3—Comfortable speed (CS) Walking (100% assisted): back-and-forth locomotion for 4 min on a 20 m length linear flat path	Trial 3—Comfortable speed (CS) Walking (100% assisted): back-and-forth locomotion for 4 min on a 20 m length linear flat path
		Trial 4—Not available	Trial 4—Comfortable speed Walking (75% assistance): back-and-forth locomotion for 4 min on a 20 m length linear flat path	Trial 4—Comfortable speed Walking (75% assistance): back-and-forth locomotion for 4 min on a 20 m length linear flat path
		Trial 5—1 min Sit-to-stand test	Trial 5—1 min Sit-to-stand test	Trial 5—1 min Sit-to-stand test
		-Task load questionnaire:	-Task load questionnaire:	-Task load questionnaire:
		NASA-TLX at the end of each Trial	NASA-TLX at the end of each Trial;	NASA-TLX at the end of each Trial
			-Usability questionnaires: SUS and TSQ-WT at the end of Session 4	-Usability questionnaires: SUS and TSQ-WT at the end of Session 5

NASA-TLX, NASA Task Load Index; Multidimensional Fatigue Inventory; SUS, System Usability Scale; TSQ-WT, Tele-healthcare Satisfaction Questionnaire—Wearable Technology.

**Table 2 sensors-25-00172-t002:** Baseline characteristics of the sample.

Subjects N°	Stroke Type and Localization	Persisting Symptoms, Complications	Additional Illness, History	Time Since Stroke (months)	FIM Score	Barthel Index Score	Muscle Strength of Lower Limbs (Evaluation by Lovett)
1	Ischemic stroke in right MCA segment	Hemiparesis, dysarthria	Hypertension II° R4 AF paroxysmal Dyslipidemia Pacemaker CABG Glaucoma	3	49	45	Right leg: 5 Left leg: 1
2	Ischemic stroke in right MCA segment	Hemiparesis, dysarthria	Hypertension II° R4 Atrial fibrillation (persistent)	2	63	40	Right leg: 5 Left leg: proximal muscle groups: 3, distal: 0
3	Hemorrhage in the right hemisphere of the brain and intraventricular hemorrhage	Hemiparesis	Hypertension II° R4 AF persistent Pyelonephritis ac. Bronchitis ac.	2,5	45	20	Right leg: 5 Left leg: proximal muscle groups: 3, distal: 2
4	Ischemic stroke in right MCA segment	Hemiparesis, dysarthria	Hypertension II° R4 Dyslipidemia	1,5	64	20	Right leg: 5 Left leg: proximal muscle groups: 3, distal: 3
5	Hemorrhage in the left cerebellum hemisphere	severe coordination disorder	Hypertension II° R4 Dyslipidemia	1	94	80	Right leg: 5 Left leg: 5

FIM—Functional independence measure, AF—atrial fibrillation, MCA—middle cerebral artery, CABG—Coronary artery bypass graft surgery.

**Table 3 sensors-25-00172-t003:** Heart rate (HR) (beats per minute—bpm) averages during every trial at every observed condition.

Subject N°	Condition 1				Condition 2					Condition 3				
	Trial1	Trial2	Trial3	Trial5	Trial1	Trial2	Trial3	Trial4	Trial5	Trial1	Trial2	Trial3	Trial4	Trial5
1	98.29	96.36	96.52	108.53	79.66	91.19	97.56	95.96	85.16	88.44	92.2	104.24	113.20	87.96
2	104.14	142.38	151.75	110.46	89.03	98.09	119.50	135.71	81.64	105.77	139.45	108.95	142.03	108.82
3	42.04	44.92	63.17	60.20	48.41	46.91	63.67	63.29	46.25	44.88	45.95	59.51	57.20	45.51
4	83.90	90.59	104.82	101.41	80.23	92.03	96.14	94.99	88.49	82.27	102.02	107.23	107.98	99.34
5	96.75	96.50	105.54	103	83.74	97.46	101.97	103.44	94.11	104.69	102.55	112.57	113.39	96.31

**Table 4 sensors-25-00172-t004:** R–R interval (milliseconds—ms) averages during every trial at every observed condition.

Subject N°	Condition 1				Condition 2					Condition 3				
	Trial1	Trial2	Trial3	Trial5	Trial1	Trial2	Trial3	Trial4	Trial5	Trial1	Trial2	Trial3	Trial4	Trial5
1	612.61	624.79	623.52	556.32	758.52	662.32	616.99	626.97	698.85	682.29	656.84	579.08	532.60	672.38
2	584.41	498.28	445.29	471.89	652.64	575.73	497.03	494.88	633.56	545.21	482.76	497.37	480.48	532.48
3	1447.17	1344	946.18	954.36	1344.82	1290.18	938.10	943.93	1253.33	1350.82	1322.11	1003.01	1044.84	1333.28
4	715.69	665.05	573.46	589.23	764.33	656.39	629.4	641.40	771.90	734.72	590.53	559.12	556.51	606.48
5	619.97	626.26	569.07	580.34	718.46	615.50	588.92	580	638.97	573.48	585.02	537.05	531.08	628.62

**Table 5 sensors-25-00172-t005:** NASA-TLX questionnaire scores.

Subject N°	Condition 1				Condition 2					Condition 3				
	Trial 1	Trial 2	Trial 3	Trial 5	Trial 1	Trial 2	Trial 3	Trial 4	Trial 5	Trial 1	Trial 2	Trial 3	Trial 4	Trial 5
1	7	15	35	35	19	47	84	92	63	15	26	65	82	69
2	6	16	24	20	6	20	59	66	31	6	18	30	36	15
3	8	15	25	27	17	29	87	98	47	7	17	40	41	29
4	6	18	25	20	6	12	11	16	13	6	12	23	23	16
5	6	16	33	35	17	48	82	95	49	6	20	54	70	59

**Table 6 sensors-25-00172-t006:** TSQ-WT questionnaire scores.

Subject N°	Condition 2	Condition 3
1	74	66
2	56	70
3	57	74
4	71	76
5	62	67

TSQ-WT, Tele-healthcare Satisfaction Questionnaire—Wearable Technology.

**Table 7 sensors-25-00172-t007:** Usability assessment with SUS.

Subject N°	Condition 2	Condition 3
1	27.5	47.5
2	42.5	50
3	25	70
4	35	40
5	37.5	47.5

SUS, System Usability Scale.

## Data Availability

The datasets presented in this article are not readily available because the data have not been published anywhere openly yet. Requests to access the datasets should be directed to raimondas.kubilius@kaunoklinikos.lt.

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
