# Peer review of "Cardiac Function and Fatigue During Exoskeleton-Assisted Sit-to-Stand Maneuver and Walking in People with Stroke with Moderate to Severe Gait Disability: A Pilot Cross-Sectional Study"

_sensors, 2024, doi:10.3390/s25010172_

Round 1
Reviewer 1 Report
Comments and Suggestions for Authors
The Authors analysed RR and HR data, together with fatigue and usability data, from 5 patients with stroke, in order to compare cardiovascular function, fatigability and workload during activities of daily living in three Conditions: 1) without any support or aid, 2) with an exoskeleton, 2) with an exoskeleton combined with a walker. The aim of the study is to assess the usability of robotic assisted gait training, as it does not compromise cardiovascular function. I think the work should go through a thorough revision process by the Authors, as some flaws need to be addressed.
1. Methods: did the Authors check for arrhythmias? Even mild arrhythmias could compromise proper measurement of HRV parameters.
2. The Authors stated “Every second subjects' HR and R-R interval were registered ...”. The reader understands that one HR value and one RR interval is registered every second. Actually, every 1 second, the Polar H9 transmits exactly one HR sample and one or more RR intervals, thus resulting in an evenly spaced HR timeseries and an unevenly spaced RR timeseries, with a number of RR samples usually greater than the number of HR samples. This is due to the fact that the device calculates RRs when beats occur and stores them in memory; then, it sends average HR (calculated using an internal circular buffer) and all the RR intervals measured in the last second over Bluetooth communication channel with a data rate of 1 packet/s. This matter should be cleared.
3. The Authors stated “Since HR and R-R interval data were normally distributed an ANOVA test was used to compare these data between all conditions (1-3) for each respected trial”. Actually, the distribution of the data to be analysed must be checked, in order to choose the appropriate statistical test for the analysis. In this case, such data is reported in Tables 3 and 4. Therefore, the Authors may want to check the distribution properties, for each trial, of the repeated measures (actually means of HR and RR values from each subject) from 5 subjects between different conditions, and not, as stated, the distribution of HR and RR series.
4. I have reproduced the analysis of TSQ-WT and SUS data in R. First, as the difference between Condition 2 and Condition 3 is distributed normally, a parametric test (one sample t test) may be used. Anyway, both parametric and non-parametric tests give non significant p-values for TSQ-WT data (p=0.21 and p=0.28, respectively). As for SUS data, both tests give p>0.05 (p=0.075 and p=0.063, respectively), while the Authors write they have found a (slightly) significant p value. Please check these calculations.
5. The most important concern is about power: all the analyses are performed on a 5-subjects sample, in a repeated measures setting. Did the Authors perform any power analysis in order to choose the appropriate sample size. Lack of effects (e.g.: no significant differences observed) may be due to lack of statistical power, and this could dramatically affect the correctness of their conclusions. The Authors correctly pointed out that the sample size is the main limitation of the study, but the conclusions may have no support from data. Therefore, asserting that “an exoskeletal robotic device can be used for gait training in people with moderate to severe disability without compromising their cardiovascular function” is hazardous. Hypothetical/conditional constructions would be more appropriate, as what is found may be suggestive of something that needs to be shown using more data.
Minors: there are several typos, please recheck the manuscript thoroughly. Some reference numbers were written with superscript characters and others not. Please uniform citation numbers to the journal’s guidelines.
Author Response
Reviewer 1
Question 1: Methods: did the Authors check for arrhythmias? Even mild arrhythmias could compromise proper measurement of HRV parameters.
Authors’ reply:
All evaluated patients were haemodynamically stable and arrhythmias and extrasystoles, which were not recorded during the study, were assessed.
Question 2: The Authors stated “Every second subjects' HR and R-R interval were registered ...”. The reader understands that one HR value and one RR interval is registered every second. Actually, every 1 second, the Polar H9 transmits exactly one HR sample and one or more RR intervals, thus resulting in an evenly spaced HR timeseries and an unevenly spaced RR timeseries, with a number of RR samples usually greater than the number of HR samples. This is due to the fact that the device calculates RRs when beats occur and stores them in memory; then, it sends average HR (calculated using an internal circular buffer) and all the RR intervals measured in the last second over Bluetooth communication channel with a data rate of 1 packet/s. This matter should be cleared.
Authors’ reply:
We appreciate the reviewer’s insightful comment, which highlighted the need for clarity regarding the data acquisition process for HR and R-R intervals using the Polar H9 device. To address this, we have revised the text in the Materials and Methods section as follows:
"Every second, the Polar H9 transmits a data packet containing one averaged HR value (derived from a circular buffer) and one or more R-R intervals measured in real time between consecutive heartbeats recorded in the previous second. This results in an evenly spaced HR time series and a non-uniformly spaced R-R interval time series.”
This updated description reflects the technical specifications of the Polar H9, ensuring the distinction between the evenly spaced HR data and the unevenly spaced R-R intervals is clear. We thank the reviewer again for pointing this out, as it has allowed us to improve the transparency and precision of our methodology.
Question 3: The Authors stated "Since HR and R-R interval data were normally distributed an ANOVA test was used to compare these data between all conditions (1-3) for each respected trial".Actually, the distribution of the data to be analysed must be checked, in order to choose the appropriate statistical test for the analysis. In this case, such data is reported in Tables 3 and 4.
Therefore, the Authors may want to check the distribution properties, for each trial, of the repeated measures (actually means of HR and RR values from each subject) from 5 subjects between different conditions, and not, as stated, the distribution of HR and RR series.
Authors’ reply:
After recent additional consultation with LUHS statistician means of HR and RR values from each subject (N=5) between different conditions during each trial were tested for distribution properties. Because the sample size is low (N=5) Shapiro-Wilk criteria was used. Requirements for normality (Gauss) distribution were met (when p>0,05 in the test):
- HR means data:
- Trial 1 under Condition 1; Trial 1 under condition 3
- Trial 2 under Condition 1; Trial 2 under condition 3
- Trial 3 under Condition 1; Trial 3 under condition 2
- Trial 4 under Condition 2; Trial 4 under condition 3
- Trial 5 under Condition 2; Trial 5 under condition 3
- RR means data:
- Trial 3 under Condition 1; Trial 3 under condition 2;
- Trial 4 under condition 2;
For comparison of data combinations mentioned in the bullets above paired samples T test was used.
Yet again, because of low sample size even for normally distributed data, to compare these data, it was recommended to use non- parametric criteria – Kruskal-Wallis.
The remaining trial and condition combinations data requirements for normality (Gauss) distribution weren’t met (p<0.05 in the test). For these data Kruskal-Wallis criteria for two dependent samples were used.
The compared combinations by tests explained above results with p<0.05:
- HR data:
Test Statisticsa |
|||
|
TR5C2 - TR5C1 |
TR5C3 - TR5C1 |
TR5C3 - TR5C2 |
Z |
-2,023b |
-2,023b |
-1,753c |
Asymp. Sig. (2-tailed) |
,043 |
,043 |
,080 |
a. Wilcoxon Signed Ranks Test |
|||
b. Based on positive ranks. |
|||
c. Based on negative ranks. |
Descriptive Statistics |
||||||||
|
N |
Mean |
Std. Deviation |
Minimum |
Maximum |
Percentiles |
||
25th |
50th (Median) |
75th |
||||||
TR5C1 |
5 |
96,7200 |
20,75698 |
60,20 |
110,46 |
80,8050 |
103,0000 |
109,4950 |
TR5C2 |
5 |
79,1300 |
18,94575 |
46,25 |
94,11 |
63,9450 |
85,1600 |
91,3000 |
TR5C3 |
5 |
87,5880 |
24,67629 |
45,51 |
108,82 |
66,7350 |
96,3100 |
104,0800 |
- RR data:
Test Statisticsa |
|||
|
TR5C2 - TR5C1 |
TR5C3 - TR5C1 |
TR5C3 - TR5C2 |
Z |
-2,023b |
-2,023b |
-1,214c |
Asymp. Sig. (2-tailed) |
,043 |
,043 |
,225 |
a. Wilcoxon Signed Ranks Test |
|||
b. Based on negative ranks. |
|||
c. Based on positive ranks. |
Descriptive Statistics |
|||||||||
|
N |
Mean |
Std. Deviation |
Minimum |
Maximum |
Percentiles |
|
||
25th |
50th (Median) |
75th |
|
||||||
TR5C1 |
5 |
630,4280 |
186,92533 |
471,89 |
954,36 |
514,1050 |
580,3400 |
771,7950 |
|
TR5C2 |
5 |
799,3220 |
259,88443 |
633,56 |
1253,33 |
636,2650 |
698,8500 |
1012,6150 |
|
TR5C3 |
5 |
754,6480 |
327,40588 |
532,48 |
1333,28 |
569,4800 |
628,6200 |
1002,8300 |
|
All the other data comparison results in p value are above 0.05.
If we may make a conclusion of all of this (according to this new data analysis by various mentioned tests above):
Data analysis suggests not only that: an exoskeletal robotic device was used without statistically significantly compromising HR and RR for these tested 5 subjects as it was mentioned in the manuscript , but also that during the trial 5 (sit-stand for a 1 minute task) subjects under con 2 (using only exoskeleton) and con 3 (using exoskeleton and UANGO walker) had less cardiovascular stress (see means in a tables above) than performing this trial under con 1(without exoskeleton).
Question 4: I have reproduced the analysis of TSQ-WT and SUS data in R. First, as the difference between Condition 2 and Condition 3 is distributed normally, a parametric test (one sample t test) may be used. Anyway, both parametric and non-parametric tests give non-significant p-values for TSQ-WT data (p=0.21 and p=0.28, respectively). As for SUS data, both tests give p>0.05 (p=0.075 and p=0.063, respectively), while the Authors write they have found a (slightly) significant p value. Please check these calculations.
Authors’ reply:
After consulting with LUHS statistician we were assured that statistically and mathematically Wilcoxon test for SUS data analysis between the groups is more accurate than your mentioned tests. And we came up to the mentioned results by using it. The main reason of Wilcoxon test selection - low number of subjects.
Wilcoxon Signed Ranks Test
Ranks |
||||
|
N |
Mean Rank |
Sum of Ranks |
|
CON3 - CON2 |
Negative Ranks |
0a |
.00 |
.00 |
Positive Ranks |
5b |
3.00 |
15.00 |
|
Ties |
0c |
|
|
|
Total |
5 |
|
|
|
a. CON3 < CON2 |
||||
b. CON3 > CON2 |
||||
c. CON3 = CON2 |
Test Statisticsa |
|
|
CON3 - CON2 |
Z |
-2.023b |
Asymp. Sig. (2-tailed) |
.043 |
a. Wilcoxon Signed Ranks Test |
|
b. Based on negative ranks. |
Question 5: The most important concern is about power: all the analyses are performed on a 5-subjects sample, in a repeated measures setting. Did the Authors perform any power analysis in order to choose the appropriate sample size. Lack of effects (e.g.: no significant differences observed) may be due to lack of statistical power, and this could dramatically affect the correctness of their conclusions. The Authors correctly pointed out that the sample size is the main limitation of the study, but the conclusions may have no support from data. Therefore, asserting that "an exoskeletal robotic device can be used for gait training in people with moderate to severe disability without compromising their cardiovascular function" is hazardous. Hypothetical/conditional constructions would be more appropriate, as what is found may be suggestive of something that needs to be shown using more data.
Authors’ reply:
We did not really have the opportunity to calculate a proper sample, as the project itself only allowed testing 5 patients with different pathologies (our centre had 5 patients with stroke). Also, the choice of five was made because it was mentioned in the discussions in various publications before the study that in the evaluation of robotic systems and exoskeletons, investigators usually use 5 subjects for pilot testing.
These are a few publications as examples, using a small sample size:
- Lamberti, Gianfranco, Gianluca Sesenna, Qamil Paja, and Gianluca Ciardi. “Rehabilitation Program for Gait Training Using UAN.GO, a Powered Exoskeleton: A Case Report.” Neurology International14, no. 2 (2022): 536–46. https://doi.org/10.3390/neurolint14020043.
- Lamberti, Gianfranco, Gianluca Sesenna, Martina Marina, Emanuela Ricci, and Gianluca Ciardi. “Robot Assisted Gait Training in a Patient with Ataxia.” Neurology International14, no. 3 (2022): 561–73. https://doi.org/10.3390/neurolint14030045.
- Sesenna, Gianluca, Cecilia Calzolari, Maria Paola Gruppi, and Gianluca Ciardi. “Walking with UAN.GO Exoskeleton: Training and Compliance in a Multiple Sclerosis Patient.” Neurology International13, no. 3 (2021): 428–38. https://doi.org/10.3390/neurolint13030042.

Reviewer 2 Report
Comments and Suggestions for Authors
1. The citation format of the references in the introduction is incorrect.
2. The number of Figure 2 is incorrect.
3. The order of Figures 1 and 2 is incorrect
4. It is necessary to analyze whether the age and degree of illness of the volunteers in the experiment have an impact on the analysis of the results.
5. Is the universality of the results based on the small number of volunteers in the experiment?
6. The paper is meaningful, but it seems more like statistics and rehabilitation assessment than scientific research.
7. The format of the references is incorrect.
Author Response
Reviewer 2
Question 1: The citation format of the references in the introduction is incorrect.
Authors’ reply: The corrections were made.
Question 2: The number of Figure 2 is incorrect.
Authors’ reply: The corrections were made.
Question 3: The order of Figures 1 and 2 is incorrect.
Authors’ reply: The corrections were made.
Question 4: It is necessary to analyse whether the age and degree of illness of the volunteers in the experiment have to impact on the analysis of the results.
Authors’ reply:
We assessed patients' comorbidities. All patients had arterial hypertension, atrial fibrillation, and dyslipidaemia (table 2), which did not significantly affect the parameters assessed in the study.
Question 5: Is the universality of the results based on the small number of volunteers in the experiment?
Authors’ reply:
We acknowledge the reviewer’s concern regarding the small sample size and its implications for the universality of our findings. We have explicitly addressed this limitation in the Discussion section of the manuscript. However, we appreciate the reviewers' insightful comments and agree that the conclusions should be tempered to reflect the exploratory nature of this pilot study. Our goal was to generate preliminary insights rather than definitive conclusions, and we acknowledge that the small sample size limits the generalizability of the results.
To address this, we have revised the Conclusions sections to adopt a more cautious tone. In particular, the statement in the conclusion paragraph:
"An exoskeletal robotic device can be used for gait training in people with moderate to severe disability without compromising their cardiovascular function."
has been rephrased to:
"The findings of this pilot study suggest that an exoskeletal robotic device may be a viable option for gait training in people with moderate to severe disability, as it does not appear to compromise cardiovascular function. However, further studies with larger sample sizes are required to confirm these preliminary results."
Question 6: The paper is meaningful, but it seems more like statistics and rehabilitation assessment than scientific research
Authors’ reply:
We sincerely thank the reviewer for their insightful comments and the opportunity to clarify certain aspects of our study.
The scientific nature of this paper is based on the hypothesis that robotic-assisted gait training has the potential to ease the process of gait training while not aggravating cardiovascular demand for people with stroke. To preliminarily verify these hypotheses, we performed a pilot cross-sectional study with randomized assessment procedures, that we considered the appropriate methodology. According to this, we designed the statistical analyses in alignment with the cross-sectional design of the study.
As for the assessment aspect, we would like to emphasize that this study was conceived as an observational investigation, designed to provide preliminary insights into the effects of exoskeleton-assisted walking on metabolic and cardiorespiratory parameters in order to inform the planning and development of larger-scale studies in the future.
Many similar publications present small samples of subjects using robotic systems and exoskeletons in medical rehabilitation. In the case of our study, we had the opportunity to test only 5 patients. However, we believe that in the long term the experience of all the researchers will be generalised and we will be able to gain more knowledge about robotic systems.
Question 7: The format of the references is incorrect.
Authors’ reply: The corrections were made.
Round 2
Reviewer 1 Report
Comments and Suggestions for Authors
All the pointed out issues have been addressed. The Authors did provide sufficient clarifications.
Reviewer 2 Report
Comments and Suggestions for Authors
none